Protogyny in a tropical damselfish: females queue for future benefit

McCormick Mark I. mark.mccormick@jcu.edu.au
ARC Centre of Excellence for Coral Reef Studies, and Department of Marine Biology and Aquaculture, James Cook University , Townsville , Queensland , Australia
Barrett Louise
Electronic publication date: 2016 Jun 30
Publication date: 2016
Volume: 4
Electronic Location ID: e2198
Received 2016 Apr 21; Accepted 2016 Jun 9
Copyright: ©2016 McCormick
Copyright year: 2016
Copyright holder: McCormick
License: This is an open access article distributed under the terms of the Creative Commons Attribution License, which permits unrestricted use, distribution, reproduction and adaptation in any medium and for any purpose provided that it is properly attributed. For attribution, the original author(s), title, publication source (PeerJ) and either DOI or URL of the article must be cited.
License URL: https://creativecommons.org/licenses/by/4.0/

Keywords: Pomacentrus amboinensis, Ambon damselfish, Behaviour, Carry-over effect, Mate competition, Dominance hierarchies, Cortisol, Parental effects, Stress, PIT tag

Funding: Australian Research Council This work was funded by the Australian Research Council. The funders had no role in study design, data collection and analysis, decision to publish, or preparation of the manuscript.

==============================
Membership of the group is a balance between the benefits associated with group living and the cost of socially constrained growth and breeding opportunities, but the costs and benefits are seldom examined. The goal of the present study was to explore the trade-offs associated with group living for a sex-changing, potentially protogynous coral reef fish, the Ambon damselfish, Pomacentrus amboinensis. Extensive sampling showed that the species exhibits resource defence polygyny, where dominant males guard a nest site that is visited by females. P. amboinensis have a longevity of about 6.5 years on the northern Great Barrier Reef. While the species can change sex consistent with being a protogynous hermaphrodite, it is unclear the extent to which the species uses this capability. Social groups are comprised of one reproductive male, 1–7 females and a number of juveniles. Females live in a linear dominance hierarchy, with the male being more aggressive to the beta-female than the alpha-female, who exhibits lower levels of ovarian cortisol. Surveys and a tagging study indicated that groups were stable for at least three months. A passive integrated transponder tag study showed that males spawn with females from their own group, but also females from neighbouring groups. In situ behavioural observations found that alpha-females have priority of access to the nest site that the male guarded, and access to higher quality foraging areas. Male removal studies suggest that the alpha-females can change sex to take over from the male when the position becomes available. Examination of otolith microstructure showed that those individuals which change sex to males have different embryonic characteristics at hatching, suggesting that success may involve a component that is parentally endowed. The relative importance of parental effects and social organisation in affecting the importance of female queuing is yet to be studied, but will likely depend on the strength of social control by the dominant members of the group.

Introduction

In organisms that live in social groups, members compete for access to limited resources and mating opportunities. Interactions among group members lead to hierarchies of resource allocation with individuals higher up in the social system having greater access to successful mating opportunities. Dominance hierarchies are the building block of stable social systems (Parker, 1974; Maynard-Smith & Parker, 1976). The social system that develops for a species and the position of individuals within the dominance hierarchy may greatly affect an individual’s life history characteristics, such as growth rate and the timing of maturation (Wong et al., 2007), and longevity (Clifton & Robertson, 1993).

When the reproductive mode of the species involves sex change, the constraints imposed by social interactions prior to and after maturation will affect growth, size-at-age, and likelihood or timing of sex change (Buston, 2003; Munday, Buston & Warner, 2006; McCormick et al., 2010). For sex changing species, individuals of the subordinate sex effectively queue for an opportunity to change to the dominant sex and maximise their fitness. To determine how social systems are maintained, it is necessary to understand how dominants optimise their negotiating power to maintain a stable social group and how subordinates optimise their own life history strategy and behaviour in response to the behaviour of dominants (Maynard-Smith & Parker, 1976; Wong et al., 2007; Buston & Zink, 2009; McCormick et al., 2010; Clutton-Brock & Huchard, 2013).

Dominants often exert control over a social group by directly or indirectly limiting growth, since in most social hierarchies rank is positively related to size or bulk (Werner & Gilliam, 1984; Drummond, 2006). Subordinates usually have lower access to foraging opportunities, restricted access to high quality food, or spend more time moving to avoid aggressive encounters. Aggressive interactions often lead to a series of endocrine responses that increase the availability of energy, partly by inhibiting processes not vital for escape or survival (Oliveira, 2004). Subordinates commonly lose interactions and this can lead to elevated glucocorticoid levels (in non-cooperative breeders; Creel, 2001) that in acute cases can lead to reproductive suppression (such as in culture), but in natural systems is more likely to lead to reduced output or offspring that are lower quality (McCormick, 1998; McCormick, 2006; Sheriff & Love, 2013). Membership of the group is then a balance between the benefits associated with group living (e.g., access to mates, lower per capita mortality; Krause & Ruxton, 2002) and the cost of socially constrained growth and breeding opportunities. However, the costs and benefits of behaviours within the context of a social system are often assumed rather than demonstrated. It is only when the benefits can be quantified that evolution of social tactics in the context of their reproductive mode can be understood.

Damselfish represent a useful model system that may provide insight into the trade-offs associated with living in a social system constrained by sexual control. Like many coral reef fishes, some damselfishes are protogynous hermaphrodites, meaning they mature as females and transition, if the social system permits, to become a dominant male (Munday, Buston & Warner, 2006; Sadovy de Mitcheson & Liu, 2008). Most damselfish species have life history attributes that are common to non-Pomacentrid fishes: an early larval phase dedicated to development and dispersal, followed by a site attached life phase devoted to growth and reproduction; a settlement transition between life stages that is central to determining their patterns of habitat distribution and density; and a strong social group that influence the trajectory of maturation and lifetime reproductive output. Unlike larger fishes, the small size of damselfishes and their site attached lifestyle permits manipulations of the social system to ascertain the factors that may determine dominance status and reproductive success and the likely costs and benefits of social status.

The goal of the present study was to explore the trade-offs associated with group living within the constraints of the social and reproductive mode they possess. This study focuses on one of the most intensively studied damselfishes, the Ambon damselfish, Pomacentrus amboinensis. This study first describes the social system and sexuality by examining its demography in an area central to the species’ distributional range. Behavioural studies of the social hierarchies within groups, hormonal assays and manipulations of group composition enable some of the costs and benefits of living in a protogynous mating system to be examined. In doing so, we gain a better understanding of the benefits of group living that lead to stable hierarchies within groups that exhibit resource defence polygyny.

Materials and Methods

Study species

Pomacentrus amboinensis, the Ambon damselfish, has a western Indo-Pacific distribution; from Indonesia to Vanuatu, north to the Ryukyu Islands, south to Scott Reef (eastern Indian Ocean) and New Caledonia. Males guard a benthic nest, often a piece of dead coral or a clam shell on a patch of sand or rubble (Fig. 1). Females lay between 1,000 and 6,000 eggs (Maddams & McCormick, 2012) on the nest before dawn (McCormick & Smith, 2004), which is guarded by the male. After a 4–5 d embryogenesis (at 28 °C) embryos hatch into ∼3 mm standard length (SL) larvae that have a pelagic larval duration of 15–23 d (Kerrigan, 1996). At the end of the larval phase fish change from being a transparent larvae to a bright yellow juvenile with an ocellus (false eyespot) and have very little morphological alteration associated with metamorphosis (McCormick, Makey & Dufour, 2002). Around the time of metamorphosis they are attracted to light at night and so are readily caught in light traps (Lönnstedt, McCormick & Chivers, 2013). The false eyespot on the posterior dorsal fin (i.e., ocellus) is flexible in size and is affected by perceived risk (Lönnstedt, McCormick & Chivers, 2013, though see Gagliano, 2008). At settlement, juveniles prefer live coral over rubble habitat when given a choice (McCormick, Moore & Munday, 2010). A combination of differential mortality associated with higher survival near territorial males (McCormick & Meekan, 2007) and interspecific competition (McCormick, 2012; McCormick & Weaver, 2012) results in juveniles being in highest abundance at the base of shallow reefs in a mixture of sand, rubble, live and dead coral. Jones (1987) examined the size distributions of male, female and immature P. amboinensis from One-Tree Island reef and concluded that the species was a protogynous hermaphrodite. Through a histological examination of the gonads Jones (1987) also found that a small number of the largest immature females were transitioning from female to males, which supported his conclusion of hermaphrodism based on the size distribution.

Figure 1 The Ambon damselfish, Pomacentrus amboinensis, have been used as a model fish for field and laboratory studies.

It is a common component of the damselfish fauna on Indo-Pacific coral reefs and is shown here with smaller P. moluccensis, a common competitor for space and resources. P. moluccensis are lemon yellow, while the three P. amboinensis photographed here have a red tinge on the dorsal surface.

Study site

All research was conducted at Lizard Island, northern Great Barrier Reef, Australia (14°40′S 145°28′E), with fish collected for demographic analysis during January and February 2001. Lizard Island is a mid-shelf island composed of granite with a well-developed fringing coral reef. P. amboinensis were collected from three ∼500 m long locations at Lizard Island: Bommie Bay on the front side of the reef, Watson’s Reef at the leeward side of the reef, and within the main lagoon at depths between 3 and 10 m. Behavioural studies and sampling of cortisol levels of individuals within groups were also undertaken from these general locations.

Fish collection

Complete social groups of P. amboinensis were randomly collected from each of the 3 locations described above. Social groups were identified by observing groups of interacting P. amboinensis for 5 min prior to sampling. Interacting groups of individuals including one or more males, various numbers of females and juveniles were found in discrete areas separated by space. The male could generally be identified (see Fig. S2) in situ and whether it was guarding a benthic nest site was recorded prior to sampling. The largest female in the area was also recorded. Individuals believed to be new recruits from the current year were not included in the sampling process (fish <∼25 mm standard length (SL)), with a minimum of 120 fish collected at each location. Individuals were captured using a dilute solution of anaesthetic clove oil and hand nets and returned to the laboratory in 5 L plastic bags of aerated seawater for processing. Once back at the laboratory, fish were sacrificed using a slurry of crushed ice and seawater. Fish did not struggle and while they would slowly stop ventilating within 30 s, they were left in the ice water at least 5 min to ensure death prior to processing. Freshly sacrificed fish were assigned a unique number, blotted dry, weighed (±0.01 g) and standard length measured with callipers (±0.01 mm). After measurement, the alimentary system with gonad tissue was removed from each fish and fixed in formaldehyde-acetic acid-calcium carbonate (FAACC; McCormick & Molony, 1992). Otoliths (sagittae) were removed under a dissecting microscope, cleaned in ethanol and stored dry storage prior to ageing. Because fish were collected in social groups, the gender, size and age composition of the social groups could be determined.

Ethics statement

This research was carried out in accordance with James Cook University ethics guidelines under ethics approval A902 and A851-03 and conducted in accordance with the Queensland Department of Primary Industries collection permit (103256) and a Great Barrier Reef Marine Park Authority research permit (G00/593). All sampling procedures and/or experimental manipulations were reviewed and approved as part of obtaining the above field and ethics permits.

Ageing

Age was estimated using the largest of the three otoliths, the sagittae. A thin transverse-section containing the nucleus of one sagittae from each fish was prepared following Wilson & McCormick (1997) and examined under immersion oil with transmitted light microscopy (×200 magnification). Annual growth rings were recorded by counting the number of opaque bands from the nucleus to the outer edge of the otolith. When the last band at the edge of the otolith was approximately half the width of the earlier band, the fish band was recorded as a half-year age. A total of 368 fish were aged for the project.

Growth bands in cross-sections of the sagittae were validated as being laid down on an annual basis using tetracycline tagging. Nineteen P. amboinensis of a range of sizes (41–61 mm SL) were collected underwater using a hand net and clove-oil, and while still at the capture site placed in a plastic bag, tagged through the bag with a fluorescent elastomer for individual identification (Hoey & McCormick, 2006), injected into the body cavity with up to 0.2 ml of a solution of tetracycline and saline (50 mg/kg body weight), kept individually in a 5 L plastic bag for 5 min to recover, and then released at the collection site. After 18 mo, all tagged fish that were still in the area were recollected using a barrier net and hand net, euthanized in a clove-oil overdose and their otoliths were removed. These otoliths were stored in the dark until being sectioned as above and viewed under ultraviolet light. Tetracycline leaves a mark on the otolith cross-sections that fluoresces under ultraviolet light. The number of light and opaque bands after the tetracycline mark were counted and this determined the periodicity of increment formation.

Historical determination of gender

Of the 368 fish whose otoliths were sectioned for age determination the clearest were further ground down to make sections ∼3–5 µm thick centred on the nucleus to enable increment counts and measurements within the larval otolith, which was bounded by the settlement mark (Wilson & McCormick, 1999). Twenty-seven (12 female and 15 male) were clear enough to see into the larval otolith and for these the diameter of the first increment was measured as this represented the size of the sagitta when the fish hatched. A t-test was used to test whether the size of the hatching otolith differed between females and those females that had changed sex to become males. There was no difference in size or age between male and female groups used in the comparison of growth histories (SL: t25 = − 2.217, p = 0.04, mean male 58.6, female 53.4 mm SL; age: t25 = − 1.272, p = 0.22, mean male 4.1, female 3.5 y; Bonferroni-adjusted alpha = 0.025).

Cost of spawning with successful males

To determine the extent to which males eat eggs from their nests the gut contents of males from nest sites monitored at 5 locations around Lizard Island (9–10 nest sites per location) were examined after nests had been monitored daily for eggs for 6-weeks during November–December 1994 (McCormick, 1998 for details). At the end of the monitoring study, fish were collected using a barrier net and a dilute clove oil solution, euthanized by clove-oil overdose, and their guts were removed and preserved in FAAC for later examination. The presence of eggs in the guts was quantified on a scale from 0 (empty) to 5 (full to capacity). The relationship of gut fullness with male reproductive success (total number of eggs present in the male nest for the monitored period) was plotted.

Social group stability

Ten groups of P. amboinensis containing at least one male, two females and a juvenile were identified at two locations (lagoon and back-reef) and individuals captured with a hand net and dilute clove-oil, measured and then tagged (underwater) for individual recognition with two colours of fluorescent elastomer (see Hoey & McCormick, 2006 for tagging methodology; see Fig. 1 for example group). Locations chosen were the edge of a continuous reef and a scaled map of each area was drawn using triangulation. Observations for 15 min were used to identify the male, α- and β-females and juveniles. Locations were visited approximately every two weeks for 3 mo and the position of each individual was plotted on the maps, together with whether they were a member of the original social group. From the maps, the mean position for each individual was determined together with the maximum distance from the mean position.

Social interactions and spacing

Fifteen social groups where the male was guarding a nest that contained eggs were examined for 20 min each and the interactions between fish within the group were tallied to determine position in the dominance hierarchy. Displays (showing side to target and erecting dorsal and anal fins), chases and bites and avoidance events were recorded. The dominant individual was judged to be the individual that had the highest score from the following calculation: displays + chases + bites—avoidances. The position of the group members in relation to the nest and one-another was recorded every 30 s. Feeding rates were also quantified for α- and β-females from 1 min focal observations. Bites were recorded regardless of whether they were successful or not.

Additional observations were made on the α-female. The number of interactions and position of the female in relation to the nest site and the male was recorded when the male was present near the nest site, and also while he was away courting other females.

Stress hormones

Because female P. amboinensis are too small to reliably remove blood from, samples of ovary tissue were collected and frozen for cortisol analysis (as per McCormick, 1998). Fish targeted for cortisol assays were the α- and β-females in the 15 chosen groups. Levels of cortisol were determined using standard radioimmunoassay (RIA) techniques (protocols see McCormick, 1998) and these were expressed as ng cortisol/g wet ovary weight.

Nest site visitation by females

Females visit the male’s nest site to gauge his success in attracting females to lay eggs and guarding the eggs once spawned for the five days (at 27 °C) of benthic development prior to hatching at 20–30 min after sunset (McCormick & Smith, 2004; Maddams & McCormick, 2012). One nest site within the shallow lagoon (2–5 m depending on tide) was chosen for an intensive study of female visitation to a male nest site, with tagging methodology detailed in McCormick & Smith (2004). Females within a ∼12 m radius of the chosen nest site were caught with a fence net, hand net and an anaesthetic clove oil solution, placed into clip-seal bags (as above), tagged in situ with fluorescent elastomer (for individual recognition, as above) and implanted into the body cavity with a passive integrated transponder (PIT) tag (TX1400BE) using a 12G needle. Females were released from the individual plastic bags at the site of capture 10 min after injection and were seen to resume feeding within 5 min of release. Females ranged in size from 39 to 62 mm SL (5–11 g). The male nests were chosen because 2-weeks of preliminary observations had found them to be receiving eggs. PIT tag readers (Destron FS2001FT) were placed over the nest site and these recorded the individual code and time stamp whenever a female with a PIT tag entered the nest site (see McCormick & Smith, 2004 for details, Fig. S3; the monitored nest site was in a similar area, but not the same nest site as previously reported). The reader was set on a 3 s delay, so that readings from the antenna (determined up to every 60 ms) were stored only every 3 s. Because the PIT tag technology uses radio waves, they get dampened after travelling only a short distance when the system is underwater seawater, hence the detector would only be triggered when a PIT tag entered the nest, and every 3 s that the tagged fish remained in the monitored nest site. Examination of the time that the antenna field was disrupted, together with the tag identity, allowed us to determine approximately how long an individual stayed within the nest site, and how this changed with female size and group membership. The nest site was monitored for 63 d. To evaluate the females’ potential contribution to eggs within the nest, the 9 d period during which eggs were present in the nest was focused on. Males do not tolerate females inside the nest unless they are spawning, and non-spawning females get quickly chased from the nest site. This observation suggests that there should be a relationship between females that spend blocks of time within the nest site and their contribution of eggs to the nest.

Male succession

My previous experiments on patch reefs involving breeding pairs of P. amboinensis have found no evidence of females changing sex when a male is present (McCormick, 1998; McCormick, 2006; Maddams & McCormick, 2012). To determine whether a female within an established social group would change sex when the male was removed, ten social groups consisting of three females and one male were established from fish that were collected from neighbouring continuous reef using a hand-net and anaesthetic clove-oil. They were placed into 5 L plastic bags of seawater, returned to the dingy, and transported over to the new lagoonal study site in a darkened 60 L tank filled with seawater to minimise any temperature change. Once at the site they were taken underwater in groups and released onto moderate sized isolated patch reefs on a sandflat located at least 20 m apart and 20 m from the reef edge. Fish were elastomer tagged underwater for individual recognition (as per Social group stability above). Reefs were comprised of live and dead Pocillopora damicornis bushy hard coral (∼1.5 × 1 m × 0.4 m) together with a plastic half tile (30 cm long, 18 cm diameter) that was used as a nesting site by the male. Once fertilized eggs were present on the nesting surface (1–2 weeks) behavioural observations allowed the determination of the dominance hierarchy among the females on the patch. Males were then removed from the patch using a fence net and the remaining social group was monitored to determine whether one of the females would take over the nest site and assume the role as a male. Recaptured males were released near their initial site of capture on the main reef (>1 km away). Monitoring was concluded once fertilized eggs were found on the nest site and these were guarded by one of the known females (or another male that migrated onto the reef).

Statistical analysis

A t-test was used to test for the difference in the diameter of the sagitta at hatching between males and females. An ANCOVA was used to test whether the maximum distance ventured from its mean position on the reef differed among four categories of social status (male, α-female, β-female and juvenile) taking into account the difference in fish size (covariate). Data were normal and homogeneous and a test for homogeneity of slopes was non-significant. A one-factor ANOVA was used to test whether social rank (1–3) of the females affected their position with respect to the edge of the reef. A paired sample t-test was used to test for equality of bite rates (per minute) for α- and β-females within 15 social groups. A paired sample t-test was also used to test for the equality of mean ovarian cortisol levels between α- and β-females for the 15 social groups for which data were available. One-factor ANOVAs were used to test for the equality of the distance to the male of the social group and proximity to the nest site between α- and β-females. For all tests assumptions were examined with residual analysis. When one-factor ANOVAs (Type III SS when unbalanced) were found to be significant the nature of significance was explored with Tukey’s HSD post-hoc tests.

Results

Size, age and gender

P. amboinensis start to mature into females at 40 mm SL, although one female was found to have oocytes at 34 mm SL (Fig. 2A). The modal size of females was 53 mm SL and the largest was 68 mm SL. Some females were still immature at 48 mm SL. Testes were found in a 30 mm SL fish, but generally the smallest individuals with testes were 40 mm SL, similar to the females. In contrast to females, the modal size of males was 60 mm SL, with the largest being 78 mm SL. A small number of non-reproductive ‘males’ (totalling 12 out of 368 fish) were found evenly spread across size classes between 38 and 68 mm SL (and across the three-week collection period). These fish had the same body morphology as mature males (Fig. S2), but had no sign of testes, even after histological inspection. These individuals may have been transitional individuals as these can have very small gonads (Sadovy de Mitcheson & Liu, 2008).

Figure 2 Population demographics.

Size (A) and age-frequency (B) distributions by gender of Pomacentrus amboinensis at Lizard Island, northern GBR. Plots show males (dark grey), females (white), juveniles (light grey) and non-reproductive males (black). N = 368 fish.

Tetracycline marking of otoliths validated that a single opaque and translucent band was laid down during one year. Eighteen months after tetracycline treatment the eight fish recollected had two opaque bands and one translucent band. The age-frequency histogram shows that the population of P. amboinensis at Lizard Island exhibits patterns of change in gender with time (Fig. 2B). As the fish age, they develop from juveniles to females to males, which is marked by the highest frequency of each sex occurring in age classes set two years apart. The highest total frequency of juveniles (29), females (54), and males (30) occupied the 2-year, 4-year, and 6-year age classes respectively. Non-reproductive males occurred evenly across age classes from 2 to 5.5 years.

Juveniles are present in the population until the age of four, while females and males do not appear in the population until the fish have reached at least the 2 y age class. Females are over twice as abundant in the population as males from the 2 y age class until their peak frequency at 4 y of age, after which they begin to decline in frequency until their maximum 5.5 y of age. Males gradually increase in frequency after the 2 y age class until their peak in the 6 y age class, where neither females nor juveniles are present.

Historical determination of gender

Measurements from cross-sections of the sagittal otolith showed that the width of the hatch mark was slightly larger in those fish that were female, than those that had changed sex to become male (t25 = − 5.432, p < 0.0001; Mean ± sd: Females, 23.53 ± 0.42; Males, 20.21 ± 0.48; Fig. S5).

Cost of spawning with successful males

Only 8 out of 47 males that guarded the monitored nests had eggs in their guts at the end of the 6-week monitoring period (Fig. S4). Egg predation was only found in males that had attracted females that spawned at least 64,000 eggs in their nest. These filial cannibals represented 8 of the 17 most successful males. Those individuals that fed on eggs had almost totally full guts.

Social group composition and dynamics

On average there was one mature male, 1.5 mature females and one immature individual within the social group (Fig. 3A). The number of females in a group ranged from none to 7, with a mode of two (Fig. 3B).

Figure 3 Social group composition.

(A) Composition of a social group, (B) frequency distribution of females within social groups around Lizard island (n = 95 social groups). Errors are standard errors.

Social groups were stable over periods of 3 mo during the breeding season at Lizard Island, with no individual from the 14 monitored groups changing membership. During censuses of mapped areas fish moved little over the monitored period, with males moving within groups the same amount as α- and β-females or juveniles (mean max distance: 1.6, 1.5, 1.5, 1.2 m respectively; ANCOVA on maximum distance moved, F3,51 = 1.853 p = 0.149). Length differences among fish of different social status did not account for a significant amount of variability in distance moved (ANCOVA, SL covariate F1,51 = 0.511, p = 0.478).

There was a negative relationship between position within the dominance hierarchy of females within the group and the size of a female relative to the α-female (Fig. 4), which accounted for 84% of the variation in dominance rank. Moreover, females of higher social rank positioned themselves further from the reef edge compared to subordinates (ANOVA, F2,27 = 39.178, p < 0.0001; Fig. 5). Alpha-females had higher bite rates than their β-females within social groups (14.6 vs 11.4 bites/min; t14 = 5.61, p < 0.0001).

Figure 4 Size affects female status.

Relationship between dominance rank of Pomacentrus amboinensis females within a social group and the size of the female relative to the alpha female. Dominance rank was determined from 15 min observation of interactions within social groups (for groups >2 females). N = 119 fish.

Figure 5 Rank affects space use.

Comparison of the distance from the reef edge versus the social rank of the females within a social group of Pomacentrus amboinensis. Errors are standard errors. Letters represent Tukey’s HSD groupings of means. N = 10 social groups.

Observations of levels of interactions among females and of females with males suggests that females live in a linear dominance hierarchy (Fig. 6). Dominant (α) females are most aggressive to β-females, show little aggression to γ-females and avoid males (F2,42 = 40.54, p < 0.0001; see Fig. 6 for Tukey’s tests). Beta-females meanwhile avoid α-females and males, but are aggressive to γ-females (F2,42 = 58.301, p < 0.0001; see Fig. 6 for Tukey’s tests).

Figure 6 Levels of aggression displayed by α-and β-female Pomacentrus amboinensis.

Data are from 20 min behavioural observations on 15 social groups. The aggression index is calculated using the formula: displays + chases + bites—avoidances. Errors are standard errors. Letters represent Tukey’s HSD groupings of means.

Alpha females had half the cortisol concentrations within their ovarian tissue than β-females (13.45 versus 28.2 ng/g; Paired t14,0.05 = − 3.497, p = 0.004), suggesting the dominant females were less stressed. Alpha females were found closer to the male than either β-females or juveniles (F2,472 = 91.054, p < 0.0001; Fig. 7A). However, β-females were further away from the nest site than either α-females or juveniles (F2,472 = 168.326, p < 0.0001; Fig. 7B).

Figure 7 Rank affects access to resources for females.

Proximity of female Pomacentrus amboinensis to the male (A) or nest site (B). Errors are standard errors. Letters represent Tukey’s HSD groupings of means. N = 8 social groups.

Males displayed to or chased β-female more than other females associated with the group or a neighbouring group, or the α-female (F3,36 = 30.613, p < 0.0001; Fig. 8). When males were absent from the nest site, such as when they were courting other females, the α-female guarded the nest, as indicated by the 8-fold increase in displays and chases by the α-female near the nest site when the male was absent (mean interactions/min ± se; male present, 0.28 ± 0.04; male absent, 2.45 ± 0.71; F1,14 = 9.452, p = 0.008).

Figure 8 Males are more aggressive to subordinate females.

Frequency of interactions by males Pomacentrus amboinensis with females or juveniles. Errors are standard errors. Letters represent Tukey’s HSD groupings of means. N = 10.

Monitoring of the visitations of PIT-tagged females through a nest site found that females were within the nest for a total of 4 h 45 min during the 9-day period (of the 63 days monitored) when eggs were present in the nest. Over this 9-day period, the nest was visited by 24 different females that lived within a 12 m radius of the nest. No one female monopolized access to the nest, with nine females accounting for 90% of the time that females spent within the nest site; the other 15 females accounted for the remaining 10%. Four of these belonged to the social group of the nest guarding male and were within the top 11 females in their total times spent within the nest (Fig. 9). There appeared to be no relationship between the total time within the nest site and female standard length.

Figure 9 Males are promiscuous.

Time spent inside the male Pomacentrus amboinensis nest by females of various sizes. The dominant female (α) and other females primarily associated with the social group are indicated (squares). Note that time is on a log10 scale. Data are from PIT tagged females entering a radio monitored nest site over a period of 9 days when eggs were present in the nest.

When males were removed from 10 social groups that had already spawned, the original α-female took over the original nest site in four instances, while in three cases a male from a neighbouring group moved to take over the nest site. In three instances there was no change in female status and no males were associated with the group during the monitoring period.

Discussion

The focal species exhibited resource defence polygyny with the male guarding a nest site that is visited by females who make the decision of whether or not to spawn with him. The composition of the group was influenced by processes internal (growth history, aggression, stress) and external (migration) to the group. The presence of a well-defined linear dominance hierarchy amongst group females means the α-females have a number of advantages including: access to food; lower stress levels (through lower male aggression); better access to the male and nest site; potentially higher quality larvae (McCormick, 1998; McCormick, 1999; McCormick, 2009); and a higher probability of taking over the nest site with male loss. In a similar way, the male benefits from the α-female through her protection of the nest when he is away from the nest site and acting as a reliable source of many, high-quality larvae, due to her large relative size and lower levels of ovarian cortisol.

Social groups were found to be spatial stable through time, which is important for the costs and benefits of being in a group to be balanced for an individual over its lifetime. In a tagging study McCormick & Makey (1997) showed that new recruits did not move more than 1.5 m in 3 mo. This study indicates that over a 3 mo period that adults and juveniles on the contiguous reef moved little from their initial locations and that they also stayed within the same social group. While many fish move from their initial nursery habitat, usually before maturity (Shulman & Ogden, 1987; Finn & Kingsford, 1996), most demersal reef-associated species have small home ranges and show high site fidelity (Marnane, 2000; Verweij & Nagelkerken, 2007; Nash et al., 2015). Stable social groups are likely to be a dominant feature of many coral reef fish species.

The size and age frequency distributions of the sexes, together with the sex change of a number of large females once the dominant males had been removed, suggest that Pomacentrus amboinensis has the capacity to change sex in a way that is characteristic of a protogynous hermaphrodite. However, in a similar way to the protandrous clownfish Amphiprion clarki, while functional sex change can occur under some conditions, the incidence in nature is unknown (Sadovy de Mitcheson & Liu, 2008). For A. clarki, dominant females lost from a group were most commonly replaced by females that migrated from outside the social group (Moyer, 1980; Ochi, 1989; Hattori & Yanagisawa, 1991). In the experimental manipulation of ten social groups, three of the social groups had males enter the group, despite being located on patch reefs on sand some 20 m from the reef edge. The males found without detectable testes in the current study may contribute to the migrants and if so may represent a viable alternative to cueing for an opportunity to change sex from a female once the position of dominant male becomes vacant. In addition, it is unclear when in life the majority sex change occurs in the species and the extent to which it is context (e.g., density or habitat) dependent. Prematurational sex change (e.g., Asoh & Kasuya, 2002 for Dascyllus trimaculatus) may occur, but further studies are required to determine the timing of sex change and the environmental or social conditions that determine the alternative sex-change pathways.

The dominance hierarchy of female P. amboinensis within the social groups was linear and based on size, with the largest individuals directing most of their aggression toward the next largest individuals. Size is a key factor in determining the outcome of many social interactions including conflict, competitive foraging and mating success (Werner & Gilliam, 1984; Warner & Schultz, 1992; Persson et al., 1996). These linear hierarchies are common amongst reef fishes (e.g., Castro & Caballero, 1998), as they are in other organisms (Clutton-Brock & Huchard, 2013), and enable the dominant individual to obtain access to resources at the expense of the submissive individual, without active aggression. The reduction or absence of aggression means unnecessary energy expenditure and the risk of injury are reduced for both individuals. In one of the few studies to attempt to determine the costs and benefits of being in a hierarchically-structured social group of tropical fishes, Clifton (1990) found that the removal of large subordinates within groups of parrotfish (Scarus iserti) caused dominant group members to increase their time spent defending territories and decreased their time spent feeding. Such social manipulations would enhance our understanding of the costs and benefits of social group membership for females in the present study species.

Dominance status, and probably larger size, had foraging benefits with males on the outer edge of the habitat further from shelter having better access to food. For the mostly planktivorous P. amboinensis (McCormick, 2003; McCormick & Weaver, 2012) individuals that are at the front of the reef are the first to receive food that arrives by currents, and may obtain higher quality food items. Forrester (1990) found that humbug damselfish (Dascyllus aruanus) that were larger occurred on the currentward edge of the reef where they obtained larger food items. This may also come at a cost with respect to survival. Clifton & Robertson (1993) found that male parrotfish had higher mortality due to their high activity and consequent exposure to predatory trevally. Currently we have no data on the relative survival rates of males compared to females or the sources of this mortality.

Benefit of subordinance is that subdominant females get the opportunity to live in a small group associated with a male who will tend their eggs. If the social group is stable, which they appear to be, then a stable hierarchy means that there are fewer challenges and more resources can be put into growth and reproduction. There are other obvious benefits of living in a diffuse group that involve reduced per capita mortality and enhanced feeding associated with the increased vigilance that comes with many eyes (Krause & Ruxton, 2002). The willingness to accept low social rank may also be a function of limited options outside the group. If resources are stable and able to be monopolised (or very variable and unpredictable), then the high competition and low probability of small individuals successfully establishing a breeding territory will favour staying within an established group (ecological constraints models, initially developed for cooperative breeding; Emlen, 1982). Subordinates may remain within a group simply because of the benefits of the protogynous social system; the more a dominant profits at the expense of subordinates, the faster it will grow and the faster it will undergo sex-change, promoting the β-fish into the α position.

There is a disjunction between the perceived stable social system and the almost open access of females to the male nest site, and this appears to decouple some of the fitness benefits for females of being a group member from the social interactions that maintain the dominance hierarchy. The high but variable egg mortality found in this species (Emslie & Jones, 2001) may mean that it is actually advantageous to have females from outside the group contributing to the egg mass on the nest site as it may effectively reduce per egg mortality rates. This is particularly the case as females can only spawn every two days (Maddams & McCormick, 2012), while males can spawn multiple times in a day. When males with egg clutches have been sacrificed at the end of our previous experiments they are often found to contain large numbers of newly spawned (<24 h post-fertilisation) eggs in their guts. This in part will be the removal of eggs that are displaying poor development. However, if females choose males to mate with based on the male’s ability to garner egg clutches and keep those through to hatching, then all the male needs to do is maintain a large clutch of eggs of a variety of developmental stages to attract females who regularly monitor the progress of nests, as found in other damselfishes (e.g., Gronell, 1989). Because maintaining and guarding eggs is energetically costly (DeMartini, 1987; Marconato, Bisazza & Fabris, 1993; Lindström, 1998), eating some of one’s own eggs (i.e., filial cannibalism, Smith & Reay, 1991) may be a cost-effective means of obtaining a high energy meal with exactly the correct nutrient profile for optimal growth. This may be beneficial to both the nest-guarding male and associated females as he may be able to put more energy into nest site defence (Lindström, 2000). It is currently unknown whether eggs that are eaten are preferentially chosen based on some assessment of quality (size, rate of development etc.), or whether clutches received from females outside the male’s group are preferentially eaten.

The finding that subordinate females have higher ovarian cortisol than α-females is in keeping with some of the literature on the endocrine correlates of dominance in fishes (e.g., Fox et al., 1997; Oeverli, Harris & Winberg, 1999). Meta-analyses of the link between stress and dominance conclude that whether or not a subordinate will express elevated cortisol will depend on the social arrangement in the vicinity of the individual (e.g., familiar individuals close by) and also whether the loser is likely to use displacement behaviour to reduce stress (Abbott et al., 2003). The most likely explanation for the present results is that most of the attention of the male and α-female is directed towards the β-female, and it is this aggression that leads to the elevated cortisol levels found in the subordinate female. One of the benefits to the α-female is that the male seldom direct aggression toward her, but rather directs it to the β-female. For the α-female, this likely results in lower levels of cortisol as a positive correlation has been found between social interactions and the levels of cortisol in the α-female of this species (McCormick, 2009).

There are many benefits for females to being large relative to other group members. Large females have higher fecundity (Maddams & McCormick, 2012; Saenz-Agudelo et al., 2015), higher dominance status and lower ovarian cortisol. Experiments on this species have shown that elevated maternal cortisol speed up the embryo developmental (McCormick & Nechaev, 2002) and results in smaller larvae at hatching (McCormick, 1998; McCormick, 1999; McCormick, 2009), which are more asymmetrical (Gagliano & McCormick, 2009) and are more likely to have lower survival during the larval phase (Lemberget & McCormick, 2009), possibly due to an altered ability to navigate (Gagliano et al., 2008).

The characteristics of the embryo appear to influence which individuals are most likely to become male. Those individuals that had transitioned to become males were found to have a smaller otolith diameter at hatching than those that were female. While the sample size is relatively small, these are the first data of their kind for a non-annual species and are supported by more substantial studies on annual protogynous fishes (Walker, Ryen & McCormick, 2007; Munday et al., 2009; McCormick et al., 2010a). McCormick et al. (2010a) found that females of haremic wrasse, Halichoeres miniatus, who changed sex to male had larger otoliths at hatching, but this was only the case when they occurred in high densities, where the social control within the group was relaxed compared to low-density groups. Little is known of what the size of the otolith at hatching means, but its relationship to larval features, such as size at hatching and yolksac size, appears to be variable and sensitive to maternal-derived cortisol in the embryo and therefore will be strongly affected by maternal effects (Gagliano & McCormick, 2009; McCormick & Gagliano, 2009). Given that males in protogynous species are the individuals with the highest lifetime fitness, this study suggests that the fitness of individuals may be, at least in part, pro-rated by characteristics of their mother during gametogenesis.

Some groups involved more than one male, but in these the subordinate male did not appear to possess detectable testes. This may occur through weighing the cost of migration and the likelihood of establishing themselves as a dominant in a group against waiting for the death of the dominant male and taking over the group, as has been shown for other animals such as the red fox (Baker et al., 1998). Because these apparently non-reproductive males are difficult to identify while the fish is alive it was not possible to determine the role these males play in the social organisation of groups.

The presence of non-spawning males may be due to the suppression of reproduction by dominant males. A number of studies have found that the presence of dominant conspecifics can suppress reproductive maturation or competence of an individual (mammals: Faulkes, Abbott & Jarvis, 1991; Bennett, Faulkes & Molteno, 1996; Saltzman, Schultz-Darken & Abbott, 1996; fishes: Leitz, 1987; Cardwell & Liley, 1991; Pankhurst & Barnett, 1993). By manipulating social groups of the cichlid Haplochromis burtoni, Fox et al. (1997) found that subordinate males had elevated cortisol levels and this lead to subordinates having smaller testes through the suppression of the reproductive hormone axis. Yet this is not always the case, and there are many examples of species in which dominant animals have higher basal glucocorticoid levels than subordinates (for review, see Sapolsky, 1982; Creel, 2001; Milla et al., 2009). In general, however, elevated cortisol levels in adult fishes do lead to reproductive suppression in either females or males (Milla et al., 2009). It is not known in the present study whether non-reproductive male P. amboinensis had higher cortisol levels compared to the reproductive dominants, but this is likely to be the case given subordinate females had higher cortisol than dominant females.

Sex change occurred at a very variable size and is likely due to the variable composition of social groups, which is a product of the vagaries of recruitment history, migration and disturbances (e.g., predation and habitat change). This suggests that the trajectory of an individual’s sexuality is controlled by factors within the group, rather than external to the group (e.g., habitat-specific growth rates). While neither females nor males were reproductively faithful to the social group they frequented, the groups were stable through time, suggesting that the advantages of group living outweighed a looser form of social organisation. Females appeared to queue for the opportunity to change sex to become the dominant male and were strongly controlled within a linear dominance hierarchy. The relaxation of social stress on the α-females may be a factor dissuading them from migrating to nearby habitat, changing sex and establishing their own social group. Dominant females have the benefit of reduced top-down social control, higher egg output (related to their size), an easily accessible nest site (and the additional foraging opportunities this may garner) and a group of familiar subordinate females that they already have a history of dominating when the opportunity arises to change sex and take over the male’s nest site. On the other hand, by having the foundation of a stable group, males gain nest site defence (from the α-female) that allows additional courting/mating opportunities, and a constant access to a large number of high quality eggs from the dominant female. While females queue for social dominance, there is the potential that greatness may be preordained because females that had successfully reached dominant-male status were also those that differed in hatching characteristics. While tantalising, the relative importance of this ‘silver-spoon effect’ compared to social queuing has yet to be determined for this or any other teleost species.

Supplemental Information

Supplemental Information 1 Supplementary materials and figures

Click here for additional data file.

We would like to thank Sheng Oon, Jennifer Drost and James Moore for assisting with the processing of the otolith samples used in this study. Shaun Smith ran the cortisol assays and showed dedication to a less than ideal PIT tagging system. B Kerrigan and numerous others assisted with the field observations and sampling. Yvonne Sadovy de Mitcheson provided very insightful comments on a draft of the manuscript.

Additional Information and Declarations

Competing Interests

Author Contributions

Animal Ethics

Field Study Permissions

Data Availability

The author declares there are no competing interests.

Mark I. McCormick conceived and designed the experiments, performed the experiments, analyzed the data, contributed reagents/materials/analysis tools, wrote the paper, prepared figures and/or tables, reviewed drafts of the paper, received grants that funded the research.

The following information was supplied relating to ethical approvals (i.e., approving body and any reference numbers):

James Cook University Animal Ethics Committee

Approvals: A902 and A851-03.

The following information was supplied relating to field study approvals (i.e., approving body and any reference numbers):

Great Barrier Reef Marine Park Authority Permit: G00/593.

The following information was supplied regarding data availability:

James Cook University Tropical Data Hub: 10.4225/28/55AED3D72AEC8

(https://research.jcu.edu.au/researchdata/default/detail/126597c430f2427c0509eb53a955eb8e/).

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
