# Peer review of "Protogyny in a tropical damselfish: females queue for future benefit"

_PeerJ, doi:10.7717/peerj.2198_

## Round 0.1 · original submission · Minor Revisions

As you will see, all our reviewers were very positive about your paper, and I agree with their assessment, and think this is a great paper for PeerJ. The reviewers do offer some constructive suggestions for improvement, however, which I would like you to consider as you prepare a final version. In particular, as reviewer 1 suggests, it would be good to make clear which data used here have been analysed and published previously in similar form, just so it's as clear as possible where the data have come from.

Congratulations on a fine piece of work, and I look forward to seeing a new version soon!

with best wishes,

Louise

Reviewer 1 ·

Basic reporting

The manuscript has been reviewed only from a scientific point of view, no language corrections have been done since the English is not my mother tongue.
In general, the manuscript has been written in a clear and unambiguous style that made it easy to read and understand; only few aspects need to be clarified (see general comments to the author). Also the structure is adequate to PeerJ standard.
The introduction is excellent and provides the proper background to get into the problem of the study. The authors also afforded many of the key references in the field.
Figures, although simple, clearly show main results of the study. Access to full data set works fine.

Experimental design

The topic of the paper perfectly fits with scope and aims of the journal. Although some of the results have been carried out in previous studies (see McCormick and Smith 2004 for nest site visitation or McCormick 1998 for stress hormones); they have been updated with new approaches and/or combine with new experiments to achieve a comprehensive description of social organization in P. amboinensis.
Experimental designs were robust a followed the ethical standard. Some of the experiments could be used to study social structure in other fish species in coastal areas and test how species-specific are these patterns.
Methods were mainly described with enough detail but some aspects need further clarification (see general comments).

Validity of the findings

The topic, social structure in sequential hermaphrodite fish species, is not particularly novel; however not many studies provide such complete set of experiments based on direct observation on how individuals interact among them within a group.
Results seem quite conclusive for most of the experiments and the statistical treatment is appropriate to the nature of experimental designs and data structure.

Additional comments

The paper is a nice compilation of field and lab experiments that provides the cues to understand the social organization and levels of hierarchy in a sequential hermaphrodite fish species. The introduction is pretty nice and easily affords the reads enough background to get into the topic of study. Updated and relevant references are cited.
I would like to say the discussion around maternal effects in later hierarchy position and sex change schedule is quite interesting and a promising field of study. Congrats.
Matherial and Methods are reasonably explained. However it is not clear enough for me which part of the field and lab work was performed ad hoc for this particular study. I say this because some of the data seems to be generated in previous published papers (see McCormick and Smith 2004 for nest site visitation or McCormick 1998 for stress hormones). I think this should be explained in a more transparent way.
I have another important concern regarding the terminology used in population demographics analysis, “immature males” or “non-reproductive males” vs. juveniles. Until I know and regarding your results, P. amboinensis is a monandric protoginy hermaphroditic species, i.e. all the individuals are females at hatching, mature as females and after, some of them, change sex to males. Regarding this theoretical sequence of maturation we cannot talk about “immature males” because individual are immature only once in life. After maturation the enter in the annual sexual cycle but they are not immature anymore (Brown-Peterson NJ, Wyanski DM, Saborido-Rey F, Macewicz BJ, Lowerre-Barbieri SK (2011) A standardized terminology for describing reproductive development in fishes. Marine and Coastal Fisheries: Dynamics, Management, and Ecosystem Science 3: 52-70). Moreover, what is the difference between “juveniles” and “immature males”? Is it possible that these “immature males” were in resting period within their annual sexual cycle? All the samples were collected during the same period of the year? Did you use histology for all the individuals? Please clarify all these aspects.
The discussion about “non-reproductive males” (ll. 505-518) is a bit speculative. Your conclusions turn around the assumption of higher levels of cortisol in non-reproductive males, but maybe -males are subject of higher levels of stress due to the nest-guarding behavior, spawning, etc. Is it possible that those smaller males act as “sneakers” during spawning? It has been demonstrated in male guppies with “sneaking” spawning strategy that they perform a better sperm quality (Evans JP (2010) Quantitative genetic evidence that males trade attractiveness for ejaculate quality in guppies. Proceedings of the Royal Society B: Biological Sciences 277: 3195-3201). Also “sneakers ” males of Coris julis in the Mediterranean produced higher testes than dominant males suggesting the adoption of a sperm competition strategy (Alonso-Fernández A, Alós J, Grau A, Domínguez-Petit R, Saborido-Rey F (2011) The use of histological techniques to study the reproductive biology of the hermaphroditic Mediterranean fishes Coris julis, Serranus scriba and Diplodus annularis. Marine and Coastal Fisheries: Dynamics, Management, and Ecosystem Science 3: 145-159).
Some specific comments are:
• Ll. 135-136: why new recruits were not included in the sampling process.
• Please provide sample size in all the sub-sections of M&M.
• Ll. 225-246: I have seen in McCormick and Smith 2004 that the detection range of the antenna is up to 6m, isn’t it? Please indicate this in the main ms. Are you assuming that each record correspond to a spawning event?
• Ll. 315-316: gut content was evaluated after the 6 week monitoring period; but, what is the rate of absorption of eggs inside the stomach? Is it possible that some individuals ate eggs at the beginning of the experiment but you did not find evidence of that at the end?
• L. 379: McCormick and Makey appears in the ms as 1998 but in the reference as 1997.

Reviewer 2 ·

Basic reporting

No comments.

Experimental design

No comments.

Validity of the findings

No comments.

Additional comments

This paper explores the trade-offs associated with living in social groups in a coral reef fish, the Ambon damselfish. Males typically guard a nest site and live in social groups with a dominant female and one or several subordinate females, juveniles and immature males. The largest female may change sex and assume the male’s role if the male disappears. Interestingly, it is suggested that maternal effects may influence the likelihood of a female changing sex to male, although at the moment the relative contribution of these putative maternal effects when compared to social controls are yet to be determined.

The manuscript is well written and easy to read. The literature review is exhaustive and very useful for any reader following the topic. The study presents a comprehensive analysis of the life-history, behaviour and social interactions of this well studied species and is innovative in determining the conditions needed for the stability of social groups that display resource defence polygyny and options available when the group’s hierarchy is altered (namely by the disappearance of the male). The range of techniques used is impressive and clearly represent an added value of this manuscript in exploring the social cues that these fish respond to.

I have very minor comments and suggestions to the manuscript and believe it is a valuable addition to the topic of research.

Lines 395-396 – It is suggested that males without visible tests could be the ones migrating to the groups after male removal/death. Additional information, if available would be important: e.g. how long does it take for such a male to start breeding? Can these males outcompete the ⍺ females?

Lines 399-402 – Some evidence that prematurational sex-change is possible should be given here

Lines 501-502 – The red fox citation is strange in this context without further explanation.

The otolith data may be added to the supplementary file showing the important back-calculated characteristics at hatching which are the basis of the speculation in the discussion of maternal effects on sex-change individuals.

Reviewer 3 ·

Basic reporting

This is an original primary research that falls well within scope of PeerJ.
The introduction and background show context, emphasizing the goal of the study: explore the trade-offs associated with group living within the constraints of their social and reproductive mode. The work is well written in clear language and the format, language and structure comply with PeerJ standard. Raw data is easily accessible in one of the links provided.

Literature is well referenced and relevant, although I suggest some recent references on social interactions, in particular those that discuss stereotypical view of sex differences (for a review see Clutton-Brock& Huchard 2013 Social competition and selection in males and females. doi:10.1098/rstb.2013.0074). This may be useful for the discussion (see more comments bellow).

Experimental design

The study focuses on a well-studied species and benefits from previous work by the same author, presenting thus a nice chain of investigation, very coherent, with all previous steps well described and referenced.
Methods are clear and described in detail.
One possible improvement is regarding the position of locations chosen for observations of social interactions. A sketch of the scaled map of each area drawn using triangulation could have been presented in a supplementary file, given its importance for the conclusions of the work.
Also, description of observations of interaction between male and females could be improved. The reader has to sort uncertainties in the text examining the figures since the text alone is somewhat ambiguous, giving the impression that observations were done only having females as focal animals (“The position of the fish in relation to the nest and the male was recorded every 30 s for each individual.”), while male behaviour was also measured.

Validity of the findings

The contribution is sound, supported by robust data, generated by careful sampling, clever experimentation and statistically sound analysis.

Some conclusions however are not fully supported by results, one of the main one being hinted in the introduction statement “For sex changing species, individuals of the subordinate sex effectively queue for an opportunity to change to the dominant sex and maximise their fitness. Along the manuscript, this does not seem to be backed up by the evidence, since there was no increased fitness (described as more foraging opportunities and access to the nest) distinguishable between the alfa female and the male. The alfa female exhibits lower levels of ovarian cortisol, has better access to food and to the nest (although Figure 9 shows that apparently access to the nest is pretty open) and is assisted by the male to keep its dominance status.

Therefore evidence presented support the conclusion that females queue up to become the dominant individual of their own sex, as this indeed would increase their fitness. Once there, the female individual will be helped by the male to keep its position and increase its fitness (by better access to foraging sites and lower stress) and thus produce better eggs that will become high-quality larvae.

In “subordinate sex effectively queue for an opportunity to change to the dominant sex” the concept of dominance is linked to aggressiveness, since dominant alfa females avoid males (but see references on stereotypical view of sex mentioned earlier). The change to the dominant- more aggressive- sex however, would not maximise their fitness, including their chance to contribute to future generations.
In addition, in the discussion it appears that “One of the benefits to the alfa female is that the male does not direct aggression toward her“, a conclusion conflicting with results that show a negative score of alfa females in relation to males
This view of dominance= aggressiveness= fitness, seems to permeate the argument, as in the discussion “Subordinates may remain within a group simply because of the benefits of the protogynous social system; the more a dominant profits at the expense of subordinates, the faster it will grow and the faster it will undergo sex-change, promoting the β-fish into the α position”. So the benefit is not only to rise to the alfa position, but to change sex. Indeed, the conclusion statement is that “Females queued for the opportunity to change sex to become the dominant male and were strongly controlled within a linear dominance hierarchy.” While the second part of the frase is indeed supported by results, the first is not.

Another important point is the so called ‘silver-spoon effect’ of a genetically preordained trend to dominance determined by maternal heritage. Indeed “Examination of otolith microstructure showed that those individuals which change sex to males have different embryonic characteristics at hatching, suggesting that success may involve a component that is maternally endowed.” However those findings showed that “The characteristics of the embryo appear to influence which individuals are most likely to become male. Those individuals that had transitioned to become males were found to have a smaller otolith diameter at hatching than those that were female.” What does a smaller otolith diameter means? If means smaller fish, it is contradictory to the expected pattern, since “Experiments on this species have shown that elevated maternal cortisol speed up the embryo developmental (McCormick & Nechaev, 2002) and results in smaller larvae at hatching (McCormick, 1998; McCormick, 1999; McCormick, 2009), which are more asymmetrical (Gagliano & McCormick, 2009) and are more likely to have lower survival during the larval phase (Lemberget & McCormick, 2009), possibly due to an altered ability to navigate (Gagliano et al., 2008).” The discussion does no seem to tackle the contradiction, why are individuals that will reach the larger sizes at maturity be the smallest larvae at hatching? And aren’t the smallest the probable offspring of lower rank females? The concluding remark does not shed any light: “Given that males in protogynous species are the individuals with the highest lifetime fitness, this study suggests that the fitness of individuals may be, at least in part, pro-rated by characteristics of their mother during gametogenesis.”

Good studies pave the way for future research so it is acceptable to leave open questions and desirable to entice the construction of new hypothesis. So further discussion should be presented around effects of growth rate and mortality and growth rate and maximum length. There is much material to inspire this discussion: Slower growing fish larvae have been shown to gave larger otoliths (since Reznick et al, 1988); determinate x indeterminate growth and larger maximum size (since Koswlovsky et al., 2004); and the large bulk of fishery papers that discuss the effect of fishing mortality on maximum size, as those can bear some interesting analogies to predation. For instance, the hypothesis of direct selection on growth rate, where fast-growing genotypes are more vulnerable to fishing could be applied to predation, where irrespective of their size because of greater appetite and correspondingly greater feeding-related activity rates and boldness that could increase mortality (see Biro & Post 2008. Rapid depletion of genotypes with fast growth and bold personality traits from harvested fish populations. doi:10.1073/pnas.0708159105).

---

## Round 0.2 · accepted · Accept

Many thanks for your revision. I found your responses clear and comprehensive, and your revisions more than appropriate. I am now very happy to accept your paper for publication, and thank you for choosing Peer J as the outlet for your work.

Once again, congratulations on a fine paper, and I hope it receives the attention it deserves.

All the best,
Lou